# Advances in Understanding the Genetic Basis of Fatty Acids Biosynthesis in Perilla: An Update

**DOI:** 10.3390/plants11091207

**Published:** 2022-04-29

**Authors:** Seon-Hwa Bae, Yedomon Ange Bovys Zoclanclounon, Thamilarasan Senthil Kumar, Jae-Hyeon Oh, Jundae Lee, Tae-Ho Kim, Ki Young Park

**Affiliations:** 1Department of Horticulture, Institute of Agricultural Science & Technology, Jeonbuk National University, Jeonju 54896, Korea; cute1004bsh@naver.com (S.-H.B.); ajfall@jbnu.ac.kr (J.L.); 2Genomics Division, National Institute of Agricultural Sciences, Rural Development Administration, Jeonju 54874, Korea; angez9914@gmail.com (Y.A.B.Z.); seninfobio@gmail.com (T.S.K.); thkim1961@korea.kr (T.-H.K.); 3R&D Coordination Division, Rural Development Administration, Jeonju 54875, Korea; jhoh8288@korea.kr; 4Department of Practical Arts Education, Gongju National University of Education, Gonju 32553, Korea

**Keywords:** fatty acid biosynthesis, *Perilla*, transcription factor, oil crop, genomics, fatty acid desaturase, triacylglycerol biosynthesis, transcriptomics

## Abstract

*Perilla*, also termed as purple mint, Chinese basil, or *Perilla* mint, is a flavoring herb widely used in East Asia. Both crude oil and essential oil are employed for consumption as well as industrial purposes. Fatty acids (FAs) biosynthesis and oil body assemblies in *Perilla* have been extensively investigated over the last three decades. Recent advances have been made in order to reveal the enzymes involved in the fatty acid biosynthesis in *Perilla*. Among those fatty acids, alpha-linolenic acid retained the attention of scientists mainly due to its medicinal and nutraceutical properties. Lipids synthesis in *Perilla* exhibited similarities with *Arabidopsis thaliana* lipids’ pathway. The homologous coding genes for polyunsaturated fatty acid desaturases, transcription factors, and major acyl-related enzymes have been found in *Perilla* via de novo transcriptome profiling, genome-wide association study, and in silico whole-genome screening. The identified genes covered de novo fatty acid synthesis, acyl-CoA dependent Kennedy pathway, acyl-CoA independent pathway, Triacylglycerols (TAGs) assembly, and acyl editing of phosphatidylcholine. In addition to the enzymes, transcription factors including *WRINKLED*, *FUSCA3*, *LEAFY COTYLEDON1*, and *ABSCISIC ACID INSENSITIVE3* have been suggested. Meanwhile, the epigenome aspect impacting the transcriptional regulation of FAs is still unclear and might require more attention from the scientific community. This review mainly outlines the identification of the key gene master players involved in *Perilla* FAs biosynthesis and TAGs assembly that have been identified in recent years. With the recent advances in genomics resources regarding this orphan crop, we provided an updated overview of the recent contributions into the comprehension of the genetic background of fatty acid biosynthesis. The provided resources can be useful for further usage in oil-bioengineering and the design of alpha-linolenic acid-boosted *Perilla genotypes* in the future.

## 1. Introduction

*Perilla frutescens* var. *frutescens* is an oil crop from the mint family that is widely distributed in East Asia including India, Vietnam, China, and Korea [1]. The *Perilla* genetic resource encompasses the oil crop type *P. frutescens* var. *frutescens*, the weedy/wild type *P. frutescens*, and wild species *Perilla setoyensis*, *Perilla hirtella*, and *Perilla citriodora* [2]. While *P. citriodora* is known as one of the diploid progenitors [3] of tetraploid *P. frutescens*, the second diploid donor has not yet been elucidated. In Korean dietary habits, *P. frutescens* var. *frutescens* is used for its oil and as leafy vegetable. The fresh leaves can serve as a wrap for meat and boiled rice and are also prepared in a pickled form [2]. In China, where it originated [1,2], *Perilla* is used secularly as a traditional herbal medicine and fragrance [2]. The health-promoting properties of this plant are attributable to its wide panel of phytochemical compounds [4]. Among them, fatty acids including omega-3, -6, and -9 have been reported as anti-cancer agents [5,6,7], coronary heart-disease protectants [8], anti-diabetic agents [9], insulin-resistant [10], anti-cardiovascular disease agents [11], and anti-depressive agents [12,13,14]. In addition, preclinical tests revealed the positive effect of *Perilla* for mitigating moderate dementia [15]. However, further investigations are required to confirm its role before a recommendation for its use as an antioxidative complement for patients with dementia [4,15]. In addition, *Perilla* is also used as a supplement in animal feeding [16,17]. Due to the numerous applications of fatty acids from *Perilla* in the health industry, the oil industry, and for animal breeding, a comprehensive background underpins fatty acid biosynthesis as a fundamental prerequisite for proper utilization in the biomedical, bioengineering, and animal industries.

Recently, *Perilla* entered into the genomics era with the sequencing of tetraploid *P. frutescence* and one diploid donor *P. citriodora* [3], laying a foundation for unraveling the genetic basis of its multiple health and nutraceutical benefits. In the present review, we will examine recent breakthroughs on the genetic basis of fatty acid biosynthesis in *Perilla*.

## 2. Earlier Identification and Cloning of Fatty Acid Encoding Gene in *Perilla*

The genetic characterization interest for *Perilla* as an oil crop with numerous health beneficial attributes started as early as the 1900s. Several fatty acid genes have been cloned and functionally characterized. Lee et al. (https://www.ncbi.nlm.nih.gov/nuccore/U59477.1/, accessed on 12 February 2021) first characterized a ω-3 fatty acid desaturase *PfrFAD7* (Genbank accession: U59477.1) extracted from a Korean cultivar “Okdong” seedling. Subsequently, a cloning of a second gene *PrFAD3* was conducted by Chung et al. [18]. *PrFAD3* exhibited a seed-specific expression when compared to other organs including the leaf, stem, and root, suggesting a preferential accumulation of alpha-linolenic acid (ALA) in the seed.

Hwang et al. [19,20] also reported four 3-ketoacyl-acyl carrier protein synthases (*KAS*) encoding genes, *PfKAS3a* (KAS III) and *PfKAS3b* (KAS III), *PfFAB1* (KAS I), and *PfFAB24* (KAS II/IV), which were responsible in the high accumulation of alpha-linolenic synthesis in *P. frutescens* seeds. Another alpha-linolenic acid-related gene, the microsomal oleate 12-desaturase (*PfFAD2*) gene, was functionally characterized for the first time in *P. frutescens* var. *frutescens* seed [21] in later studies. In addition to the previously identified *FAD3* and *FAD7* type genes, Xue et al. [22] isolated two *FAD8* alpha-linoleic-related genes (*PrFAD8a* and *PrFADb*) harboring two pyrimidine stretches. Interestingly, the expression of *PrFAD8* genes was predominantly observed in the *Perilla* bud while its accumulation increased under injury, Methyl jasmonate (MeJA), Salicylic Acid (SA), and Abscisic acid (ABA) effects; highlighting their implications in plant defense, growth, and development.

## 3. Transcriptomics Sheds Lights into Key Master Player Enzymes of *Perilla* Fatty Acid Biosynthesis

Although some genes have been investigated earlier, the fully resolved biosynthesis pathway of fatty acids in *Perilla* was still unclear. To fill this gap, the RNA sequencing approach has been extensively used because it helps in uncovering expressed genes related to a biological process. By deciphering the transcriptome of *Perilla* using diverse organs, scientists were able to identify key genes related to fatty acid biosynthesis via de novo transcripts assembly and functional gene prediction. Thus, extensive transcriptome studies have been initiated using different materials, including *P. fruescens* var. *frutescens*, *Perilla frutescens* var. *crispa f. purpurea* (red *Perilla*), and *P. frutescens* var. *crispa f. viridis* (green *Perilla*) [23,24,25,26]. The uncovered key genes involved in fatty acid biosynthesis in *Perilla* have been summarized in Figure 1. Briefly, based on *Perilla’s* fatty acid desaturase subcellular localization prediction [27] and the well-studied Arabidopsis fatty acid biosynthesis model [28], most fatty acids, including palmitic acid (C16:0), stearic acid (C18:0), and oleic acid (C18:1), were exclusively synthesized in plastids and conveyed into the cytoplasm where they entered into an acyl-CoA pool for the esterification process at sn-2 position resulting in phosphatidylcholine under the acyl-CoA:lysophosphatidylcholine acyltransferase (*LPCAT*) enzyme effect.

Oleic acid was then desaturated in the endoplastic rediculum (ER) to become consecutively linoleic acid (LA) and alpha-linolenic acid (ALA) under *FAD2* and *FAD3* genes, respectively. The resulting polyunsaturated fatty acids were transacylated onto the sn-3 position of diacylglycerol by phospholipid:diacylglycerol acyltransferase (*PDAT*) or returned to the acyl-CoA pool via *LPCAT* to be incorporated into TAG through the Kennedy pathway, inducing the production of triacylglycerols (TAGs) [29].

Using *Perilla* as a plant model, numerous fatty acid-related genes have been identified. From a time-course seed transcriptome analysis, Kim et al. [25] identified 43 acyl-lipid related genes in *P. frutescens* var. *frutescens* cv. *Dayudeulkkae* (Table 1). The identified genes via Arabidopsis orthologs detection covered the de novo fatty acid biosynthetic key enzymes present in the plastid, endoplasmic reticulum desaturases, oil body proteins, acyl-CoA-, and phosphatidylcholine-mediated TAG synthesis.

Transcriptome mining revealed five sub-unit genes (*α-PDH*, *β-PDH*, *EMB3003*, *LTA2*, and *LPD1*) of the precursor enzyme plastidial pyruvate dehydrogenase complex (*PDHC*) involved in the synthesis of acetyl-CoA from pyruvate. Afterward, acetyl-CoA carboxylase (*ACCase*) transformed acetyl-CoA ito malonyl-CoA [30]. The *ACCase* in *Perilla* encompassed two ACCases subunits alpha (*α-CTa* and *α-CTb*), one ACCase subunit beta (*β-CT*), two isoforms of biotin carboxyl-carrier protein (*BCCP1* and *BCCP2*), and one biotin carboxylase (BC).

Furthermore, the malonyl-CoA ACP transacylase, an acyl carrier protein transacylase, catalyzed malonyl-CoA to form malonyl-ACP, paving the way for fatty acid elongation under the action of acyl-chain enzymes, i.e., 3-keto-acyl-ACP synthase (*KAS*), 3-ketoacyl-ACP reductase (*KAR*), 3-hydroxylacyl-ACP dehydratase (*HAD*), and Trans-∆2-enoyl-ACP reductase (*EAR*), respectively [23,24,31]. It is worth mentioning that *WR1* is well conserved in plant species. For instance, homologous genes have been identified in *Brachypodium distachyon* [32], *Camelina sativa* [33], *Solanum tuberosum* [34], *Cocos nucifera* [35], *Brassica napus* [36], *Elaeis guineensis* [37], and *Jatropha curcas* [38]. In *A. thaliana*, through the promoter binding element AW-box, *WRI1* targets upstream genes encoding for malonyl-CoA:ACP malonyl transferase, enoyl-ACP reductase, pyruvate dehydrogenase, oleoyl-ACP thioesterase, biotin carboxyl carrier protein 2, ketoacyl-ACP synthase, and hydroxyacyl-ACP dehydrase [39,40,41,42,43,44,45,46]. The homologous sequence of *WR1* has been demonstrated in augmentation from 10 to 40% of seed oil in transgenic maize [47] and *Brassica napus* [36], suggesting that *Perilla’s WR1* gene might be a promising candidate for oil-oriented bioengineering in *Perilla*.

Through carbon chain elongation, palmitoyl-ACP (C16:0) is converted into stearoyl-ACP (C18:0). The latter is transformed into oleic acid (C18:1)-ACP under the catalysis of stearoyl-acyl carrier protein desaturase (*SAD*). In *Perilla*, two *SAD* genes have been identified, including *PfFAB2* and *PfDES6* [25]. Using a red *Perilla* (*Perilla frutescens* var. *crispa F. purpurea*) seed transcriptome, Liao et al. [23] identified fatty acid desaturases PfFAD6 and PfFAD7/8 that act on the vector glycerolipid, i.e., monogalactosyldiacylglycerol (MGDG), in order to process (C18:1) into (C18:2) and (C18:2) to (C18:3), respectively (Figure 1).

To terminate fatty acids synthesis in *Perilla* plastids, fatty acyl-ACP thioesterase (*FATA*), palmitoyl/stearoyl-acyl carrier protein thioesterase (*FATB*), and palmitoyl-CoA hydrolase (*PCH*) were solicitated. *PCH* specifically induced C18:1- and C18:2-synthesis, while FATA was a C18:1-exclusive catalyst. Meanwhile, *FATB* transformed only C16:0-ACP or C18:0-ACP to C16:0 or C18:0, respectively (Figure 1). Representative gene coding for these enzyme has been pinpointed by de novo transcriptome analysis and comparative transcripts with regard to the well characterized *A. thaliana* fatty acid-related gene [23,24]. Free FAs were then moved into the cytoplasm where they were esterified to form an Acyl-CoA pool under the action of long-chain acyl-COA synthesis (*LACS*). Liao et al. [23] reported the important expression of *LACS* genes in *Perilla* seeds ten days after flowering, indicating an initiation of TAGs synthesis pathway in the endoplasmic reticulum (ER).

In the ER, esterified fatty acids are translated into phosphatidylcholines via lysophosphatidylcholine acyltransferase (*LPCAT*). Based on the Arabidopsis plant model, mainly two fatty acid desaturases have been identified in the ER: an *FAD2* that converts PC-C18:1 into PC-18:2 and an *FAD3* that catalyzes PC-C18:2 into PC-C18:3 [48,49,50]. Homologous sequences in *Perilla* seed (*PfFAD2* and *PfFAD3*) transcriptome [23,24,25] have also been identified (Table 1).

Recently, the transcriptome assessment of Chinese cultivar PF40 highlighted 33 candidate genes involved in TAG biosynthesis-covering transcription factors (Appendix A), and fatty acids were exported from plastid, acyl editing of phospatidylcholine, acyl-CoA dependent Kennedy pathway, acyl-CoA independent pathway, and TAGs assembly into oil bodies (Table 1). The identified genes corroborated with previous findings [23,24,25], except for the first identification of fatty export1 (*FAX1*) as an additional enzyme to long-chain acyl-CoA synthetase (*LACS*) that mediated plastid fatty acid export.

In the absence of a whole genome representative resources, the detection of potential genes isoforms and the full FADs gene repertoire is difficult to predict, and diverse gene targets for functional validation and bio-engineering purposes are not provided. Due to the fact that *Perilla* has entered into the genomics era, the next section covers genomics-based advances in the detection of fatty acids in *Perilla* via genome-wide identification and genome-wide association study strategies.

## 4. Whole-Genome-Driven Fatty Acid Genes Discovery

With the advent of long-reads and chromosome conformation capture technologies, a high-quality chromosome scale genome of tetraploid *P. frutescens* var. *frutescens* has recently been assembled [3]. The genome spanned 1.203 Gb, along with 20 chromosomes with an N50 of 62.64 Mb and a total of 38,941 predicted gene models.

From a panel of 191 accessions, a genome-wide association study for seed alpha-linolenic acid content enabled the identification of an *LPCAT* encoding region located in chromosome 2. This finding corroborates previous observations, suggesting the role of *LPCAT* in FAs and TAGs synthesis in *B. napus* [51] and *A. thaliana* [52]. Interestingly, a deletion of this gene was noted in some individuals of the studied panel corresponding to a loss of around 6% of seed oil ALA content. This suggests that the transcriptional regulation of *LPCAT* might be responsible for ALA content variations in *Perilla*.

Taking advantage of the PF40-generated high-quality genome, in silico genome-wide analysis identified a repertoire of 42 fatty acid desaturases clustered into five families including *omega-3 desaturase*, *∆7/**∆9 desaturase*, *FAD4 desaturase*, *∆12 desaturase*, and *front-end desaturase* [27]. The heterologous validation of candidate fatty acid desaturase genes using *A. thaliana* revealed a positive impact (increase of 18–37% alpha-linolenic acid content) of the *PfFAD3.1* gene.

Furthermore, the upregulation of *WRINKLED* (*WRI1*), *FUSCA3* (*FUS3*), *LEAFY COTYLEDON1* (*LEC1* and *LCE2*), and *ABSCISIC ACID INSENSITIVE3* (*ABI3*) transcription factors was noted in *PfFAD3.1* Arabidopsis transgenic lines [3] and Perilla seed expression profiles [23], suggesting their regulation roles in the *Perilla* FAs synthesis pathway.

## 5. Concluding Remarks and Outlook

Fatty acids play an important role in the lipid supply of plants and have valuable medicinal properties for humans. Here, we summarized the breakthroughs that shed light into the genetic and molecular determinants of FA and TAG synthesis in *Perilla*. Transcriptomics and genomics studies revealed the key master player enzymes responsible for FAs synthesis in *Perilla,* including polyunsaturated fatty acids desaturases, acyl-related enzymes, and transcription factors. However, the evidence of their role is still elusive since strong functional validation has not yet been provided.

The mechanism of the regulation of FA synthesis by TFs in *Perilla* is still elusive. Meanwhile, the recent work from Moreno-Perez et al. [53] suggested histone methylation (H3K4me3) implication into fatty acid biosynthesis in sunflowers with interactions with TFs. Moreover, acetyl-CoA, which is involved in fatty acid synthesis in plants, has been found to be correlated with histone acetylation and DNA methylation in *A. thaliana* through the beta-oxidation process [54]. Therefore, an in-depth investigation of identified TFs, such as *ABI3*, *FUS3*, *LEC1*, and *LEC2*, and the epigenome landmark of *Perilla* will pave a new avenue in deciphering the full landscape of fatty-acid biosynthesis in *Perilla*.

Functional validation using *Perilla* as a material instead of *A. thaliana* would drastically shape the validation efficiency of the identified genes. For this purpose, Agrobacterium-based protocols [55,56] have been tested and can serve as further functional validation. Moreover, in the current era of gene and genome editing with applicable cases in plants [57,58,59,60], designing appropriate gene editing strategies that fit into the *Perilla* system will surely expedite the production of enriched alpha-linolenic acid-*Perilla* genotypes. Furthermore, considering the species diversity within the *Perilla* genus, systematic fatty acid content evaluation within the *Perilla* species will help reveal potential alpha-linolenic acid-enriched species donors and characterize their respective biosynthetic pathways.

## Figures and Tables

**Figure 1 plants-11-01207-f001:**
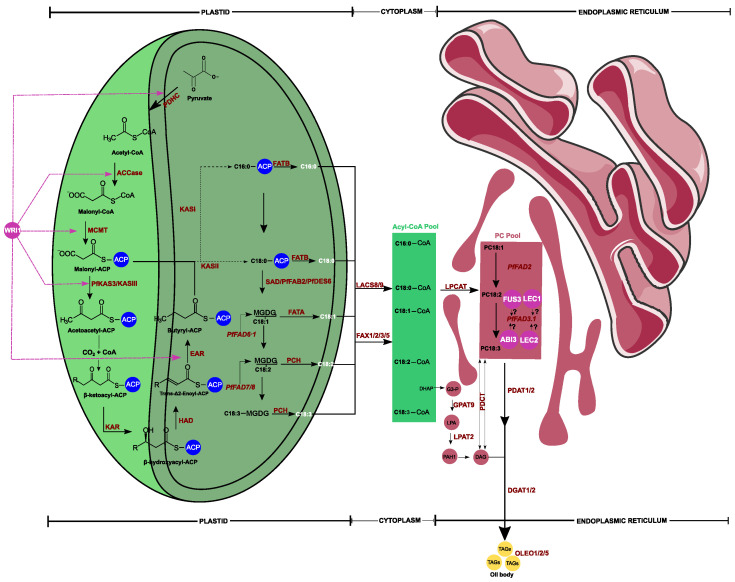
A simplified putative diagram view of fatty acids biosynthetic pathway in *Perilla* and triacylglycerols (TAGs) assembly. The schematic view involved bio-chemical interactions occurring in plastid, cytoplasm, and endoplasmic reticulum, respectively. The resulting TAGs are indicated in yellow. Purple circles indicate transcription factors, including WRINKLED (*WRI1*), FUSCA3 (*FUS3*), LEAFY COTYLEDON1 (*LEC1*, *LCE2*), and ABSCISIC ACID INSENSITIVE3 (*ABI3*). The transcriptional regulation of *FUS3*, *LCE1*, *LCE2*, and *ABI3* with *PfFAD3.1* is not yet uncovered. *PDHC*: plastidial pyruvate dehydrogenase complex; *ACCase*: acetyl-CoA carboxylase; *MCMT*: malonyl-CoA ACP transacylase; *KASIII*: ketoacyl-ACP synthase type III; *KAR*: 3-ketoacyl-ACP reductase; *HAD*: 3-hydroxyacyl-ACP dyhydratase; *EAR*: 2-enoyl-ACP reductase; *KASII*: ketoacyl-ACP synthase type II; *KASI*: ketoacyl-ACP synthase type I; *SAD*: stearoyl-acyl carrier protein desaturase; *FATB*: acyl-ACP thioesterase B; *FATA*: acyl-ACP thioesterase A; MGDG: monogalactosyldiacylglycerol; *PfFAD*: *Perilla frutescens* fatty acid desaturase; PC Pool: phosphatidylcholines pool; *PCH*: palmitoyl-CoA hydrolase; *LACS*: long-chain acyl-CoA synthetase; *PDCT*: phosphatidylcholinediacylglycerol cholinephosphotransferase; *FAX*: fatty acid export; *LPCAT*: lysophosphatidylcholine acyltransferase; *PDAT*: phospholipid diacylglycerol acyltransferase; *DGAT*: diacylglycerolacyltransferase; *GPAT*: glycerol-3-phosphate acyltransferase; *LPAT*: 1-acylglycerol-3-phosphate acyltransferase; *DHAP*: dihydroxyacetone phosphate; *PAH*: phosphatidic acid phosphatase; *OLEO*: Oleosin.

**Table 1 plants-11-01207-t001:** Summary of Identified Major Genes Involved in Fatty Acid and Triacylglycerols Biosynthesis in *Perilla*.

Enzyme ID	Enzyme Name	GeneID			Homologous	Pathways Involved	Field of Study	References
		PF40 *	Dayudeulkkae **	PC ***	*A. Thaliana*			
*PDH(E1α)*	Pyruvate Dehydrogenase E1 Subunit Alpha 1		Locus_2112		AT1G01090.1	FA de novo biosynthesis and export from plastid	Transcriptomics	[25]
*PDH(E1ß)*	Pyruvate Dehydrogenase E1 Subunit beta 1		Locus_25208		AT2G34590.1	FA de novo biosynthesis and export from plastid	Transcriptomics	[25]
*EMB3003(E2)*	Pyruvate dehydrogenase e2 component (dihydrolipoamide acetyltransferase)		Locus_33306		AT1G34430.1	FA de novo biosynthesis and export from plastid	Transcriptomics	[25]
*LTA2 (E2)*	Plastid E2 Subunit of Pyruvate Decarboxylase, PLE2		Locus_5104		AT3G25860.1	FA de novo biosynthesis and export from plastid	Transcriptomics	[25]
*LPD1 (E3)*	Lipoamide dehydrogenase		Locus_7407		AT3G16950.1	FA de novo biosynthesis and export from plastid	Transcriptomics	[25]
*α-CTa*	Alpha-carboxyltransferase Isoform a		Locus_8492		AT2G38040.1	FA de novo biosynthesis and export from plastid	Transcriptomics	[25]
*α-CTb*	Apha-carboxyltransferase Isoform b		Locus_2178		AT2G38040.1	FA de novo biosynthesis and export from plastid	Transcriptomics	[25]
*ß-CT*	Beta-carboxyltransferase		Locus_53041		ATCG00500.1	FA de novo biosynthesis and export from plastid	Transcriptomics	[25]
*BC*	Biotin carboxylase		Locus_22078		AT5G35360.1	FA de novo biosynthesis and export from plastid	Transcriptomics	[25]
*BCCP1*	Biotin carboxyl carrier protein of acetyl-CoA carboxylase 1		Locus_29162		AT5G16390.1	FA de novo biosynthesis and export from plastid	Transcriptomics	[25]
*BCCP2*	Biotin carboxyl carrier protein of acetyl-CoA carboxylase 2		Locus_17340		AT5G15530.1	FA de novo biosynthesis and export from plastid	Transcriptomics	[25]
*MCMT*	Malonyl-CoA ACP transacylase		Locus_14579		AT2G30200.1	FA de novo biosynthesis and export from plastid	Transcriptomics	[25]
*KASIII*	3-Ketoacyl-ACP synthase		Locus_10821		AT1G62640.1	FA de novo biosynthesis and export from plastid	Transcriptomics	[25]
*KAR*	3-ketoacyl-ACP reductase		Locus_1445		AT1G24360.1	FA de novo biosynthesis and export from plastid	Transcriptomics	[25]
*HAD*	3-hydroxyacyl-ACP dyhydratase		Locus_19332		AT5G10160.1	FA de novo biosynthesis and export from plastid	Transcriptomics	[25]
*EAR*	2-enoyl-ACP reductase		Locus_25443		AT2G05990.1	FA de novo biosynthesis and export from plastid	Transcriptomics	[25]
*FATA*	Fatty acyl-ACP thioesterase A		Locus_29919		AT3G25110.1	FA de novo biosynthesis and export from plastid	Transcriptomics	[25]
*FATB*	Fatty acyl-ACP thioesterase B		Locus_6603		AT1G08510.1	FA de novo biosynthesis and export from plastid	Transcriptomics	[25]
*FAB2*	Fatty acid biosynthesis2		Locus_13564		AT2G43710.1	FA de novo biosynthesis and export from plastid	Transcriptomics	[25]
*DES6*	Stearoyl-acyl carrier protein desaturase		Locus_9486		AT1G43800.1	FA de novo biosynthesis and export from plastid	Transcriptomics	[25]
*KASI*	Ketoacyl-ACP Synthase I		Locus_26341		AT5G46290.1	FA de novo biosynthesis and export from plastid	Transcriptomics	[25]
*KASII*	Ketoacyl-ACP Synthase II		Locus_1373		AT1G74960.1	FA de novo biosynthesis and export from plastid	Transcriptomics	[25]
*LACS8*	Long-chain acyl-CoA synthetase 8	chr07_36292788_36299197 chr19_22302145_22308533	Locus_3838	chr06_37084362_37090768	AT2G04350.1	FA de novo biosynthesis and export from plastid	Genome Assembly, Transcriptomics	[3,25]
*LACS9*	Long-chain acyl-CoA synthetase 9	chr03_70622879_70627324 chr09_58852417_58856892 chr01_02424545_02428997	Locus_23636	chr01_02424545_02428997	AT1G77590.1	FA de novo biosynthesis and export from plastid	Genome Assembly, Transcriptomics	[3,25]
*FAX1*	Fatty acid export 1	chr05_24282740_24284950 chr01_71691539_71693779		chr02_42552603_42554830		FA de novo biosynthesis and export from plastid		[25]
*FAX2*	Fatty acid export 2	chr07_10626150_10628000		chr06_11381976_11383822		FA de novo biosynthesis and export from plastid		[25]
*FAX3*	Fatty acid export 3	chr04_00857340_00859552		chr03_67540865_67543081		FA de novo biosynthesis and export from plastid		[25]
*FAX5*	Fatty acid export 5	chr04_65527957_65529911 chr07_22534802_22537586 chr06_00746938_00748860 chr19_10735560_10738363		chr03_02347871_02349825 chr06_23562111_23564893		FA de novo biosynthesis and export from plastid		[25]
*FAD2*	Omega-6 fatty acid desaturase	chr12_56933298_56934446 chr11_05592060_05593208 chr11_05575254_05576393	Locus_733	chr08_55538081_55539229	AT3G12120.1	Acyl editing of phospatidylcholine	Genome Assembly, Transcriptomics	[3,25]
		chr12_56948107_56949167		chr08_55558209_55559348				
*FAD3*	Omega-3 fatty acid desaturase	chr12_04645208_04647776 chr11_54194712_54197265	Locus_22029	chr08_04030082_04032640	AT2G29980.1	Acyl editing of phospatidylcholine	Genome Assembly, Transcriptomics	[3,25]
*FAD8*	Omega-8 fatty acid desaturase		Locus_5107		AT5G05580.2	Acyl editing of phospatidylcholine	Transcriptomics	[25]
*GPAT9*	Glycerol-3-phosphate acyltransferase 9	chr12_33733527_33737891 chr11_26255533_26259881	Locus_10180	chr08_33038421_33042132	AT5G60620.1	Acyl-CoA-dependent TAG synthesis in Kennedy pathway	Genome Assembly, Transcriptomics	[3,25]
*LPAT2*	1-acyl-sn-glycerol-3-phosphate acyltransferase 2	chr05_23583386_23588593 chr05_34400913_34404444 chr01_72114246_72119454	Locus_6587	chr02_43313059_43318262 chr02_32585727_32589258	AT3G57650.1	Acyl-CoA-dependent TAG synthesis in Kennedy pathway	Genome Assembly, Transcriptomics	[3,25]
*PAH1*	Phenylalanine hydrolase 1	chr01_61567423_61570965 chr14_08597119_08602056 chr15_37103964_37108907 chr03_61656532_61661875 chr18_09154357_09159306 chr17_34575710_34580664 chr09_50343045_50349360		chr10_43830659_43835596 chr01_11516392_11522733		Acyl-CoA-dependent TAG synthesis in Kennedy pathway		
*DGAT1*	Diacylglycerol O-acyltransferase 1	chr01_09730655_09741367 chr01_48275733_48286173	Locus_14696	chr05_08797620_08808333	AT2G19450.1	Acyl-CoA-dependent TAG synthesis in Kennedy pathway	Genome Assembly, Transcriptomics	[3,25]
*DGAT2*	Diacylglycerol O-acyltransferase 2	chr14_26782964_26787941 chr18_25811826_25816791	Locus_12629	chr10_25785382_25790335	AT3G51520.1	Acyl-CoA-dependent TAG synthesis in Kennedy pathway	Genome Assembly, Transcriptomics	[3,25]
*DGAT3*	Diacylglycerol O-acyltransferase 3		Locus_1560		AT1G48300.1	Acyl-CoA-dependent TAG synthesis in Kennedy pathway	Transcriptomics	[25]
*LPCAT*	Lysophosphatidylcholine acyltransferase	chr01_06996630_07001595 chr05_56678891_56685081 chr01_03079195_03084058 chr07_53028425_53034567 chr01_43224061_43229071 chr02_66141068_66147271 chr02_04634020_04638876 chr19_35211932_35217537	Locus_43749	PC00000058_00436672_00441634 chr02_10454190_10460391 chr05_03185967_03190829 chr06_54113419_54119561	AT1G12640.1	PC-mediated TAG synthesis	Transcriptomics	[3,25]
*CPT1*	Diacylglycerol cholinephosphotransferase		Locus_7821		AT1G13560.1	PC-mediated TAG synthesis	Transcriptomics	[25]
*CPT2*	Diacylglycerol cholinephosphotransferase		Locus_22567		AT3G25585.1	PC-mediated TAG synthesis	Transcriptomics	[25]
*PDAT1*	Phospholipid:diacylglycerol acyltransferase 1	chr05_44104376_44108847	Locus_7255	chr02_22969948_22974420	AT5G13640.1	Acyl-CoA independent pathway	Transcriptomics	[3,25]
		chr03_00447151_00451507		PC00002899_00154872_00159184				
		chr02_52135886_52140327						
		chr09_00376677_00380564						
*PDAT2*	Phospholipid:diacylglycerol acyltransferase 2	chr05_38922115_38924735	Locus_29208	chr02_28050267_28052887	AT3G44830.1	Acyl-CoA independent pathway	Transcriptomics	[3,25]
		chr02_45992086_45994691						
*PDCT*	Phosphatidylcholine:diacylglycerol cholinephosphotransferase	chr03_46291224_46293449 chr09_37050943_37053194	Locus_15867	chr01_27228085_27230144	AT3G15820.1	Acyl-CoA independent pathway	Genome Assembly, Transcriptomics	[3,25]
*OLEO2*	Oleosin2	chr15_52133834_52134256	Locus_31790		AT5G40420.1	TAG assembly	Transcriptomics	[3,25]
		chr17_50355018_50355440		chr09_02008310_02008732				
*OLEO*	Oleosin	chr14_08347244_08347714	Locus_31788	chr10_44101965_44102435	AT3G18570.1	TAG assembly	Transcriptomics	[3,25]
		chr18_08871500_08871970						
*OLEO1*	Oleosin1	chr05_05196095_05196523	Locus_29266	chr02_64426568_64426996	AT4G25140.1	TAG assembly	Transcriptomics	[3,25]
		chr01_30156121_30156549						
*OLEO5*	Oleosin5	chr05_59989345_59989911	Locus_29276		AT3G01570.1	TAG assembly	Transcriptomics	[3,25]
		chr05_59997449_59997976		chr02_07157257_07157823				
		chr02_69562819_69563393		chr02_07149192_07149719				
		chr02_69577662_69578195						

* *Perilla frutescens* var. *frutescens* cv. PF40; ** *Perilla frutescens* var. *frutescens* cv. Dayudeulkkae; *** *Perilla citriodora*. The mentioned genes have been identified through de novo transcriptome mining coupled with Arabidopsis homologous sequences prediction.

## Data Availability

The data presented in this study are available in Appendix A.

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
