# Peer review of "Advances in Understanding the Genetic Basis of Fatty Acids Biosynthesis in Perilla: An Update"

_plants, 2022, doi:10.3390/plants11091207_

Round 1
Reviewer 1 Report
It would be really helpful if the authors make sure that they specify in their statements whether the findings are coming from Arabidopsis or other species or have been clearly established in Perilla and in which tissues, seeds, leaves etc… This will help the reader to clearly identify what is known in Perilla.
Moreover, some statements lack accuracy.
Please, find my comment in the pdf file attached. The comments can be seen when using Abode Acrobat reader (free version).

Reviewer 2 Report
The manuscript entitled " Advances in Understanding the Genetic Basis of Fatty Acids Biosynthesis in Perilla: An update" discussed recent genetic developments in understanding PUFA rich oil biosynthesis in Perilla. The article is very narrow and will not add any significant information to the existing knowledge. The article needs to be improved by discussing fatty acid content changes and the genetic divergence of fatty acid genes between perilla species.
Reviewer 3 Report
The manuscript by Bae et al. is a nice summary for genetic basis of fatty acid synthesis in Perilla. The manuscript is organized into sections that make it easier for the readers to digest. There is an elaborate list of genes in fatty acid biosynthesis in table 1. That being said, there are some typos that will unfortunately lessen the impression of the quality readers would have of the article. I would encourage the authors to have the manuscript extensively reviewed and proofread before re-submission. I also have other minor suggestions that need to be addressed.
1) check abstract: WRINKLED, not WRINCKLED
2) check abstract: phosphatidylcholine, not phosphatydilcholine
3) Please make sure to use ACCase throughout, not Accase
4) the authors mentioned master regulators of fatty acid biosynthesis in the abstract but did not go into further detail. Including the latest studies on these master regulators would strengthen the manuscripts.
5) the authors summarized fatty acid synthesis pathway and briefly mention that they are under regulations of transcription factors, for example, WRI1. Recent advances in fatty acid biosynthesis also showed upstream proteins that decide the fates of WRI1, to be degraded or to be stabilized. The authors did not mention any of these upstream proteins that interact with master regulators mentioned in the abstract at all.
6) There should be a separate table for transcription factors.
7) The authors included WRI in table 1 but left out all other transcription factors.
Round 2
Reviewer 2 Report
The authors have not improved the manuscript compared to the previous version. Authors should discuss challenges and future perspectives in understanding the genetic basis of the crop with respect to improving fatty acid quality and quantity.
Author Response
Dear Reviewer,
Thank you to raise our attention regarding the challenges and perspectives. In the present manuscript, we raised the point of weakness and advances in the Perilla case. Since all the studies are based on transcriptome investigation and homology-based orthologs detection with A. thaliana as reference. We also proposed that functional validation has to be initiated to confirm the role of the key enzymes. We also pointed out the potential role of transcription factors, mainly WRI1, in fatty acid regulation. We suggested the investigation of the other Perilla species' wild relatives to find out novel genes since wild resources encompass useful genes that can be relevant for a specific trait targeting. All those recommendations have been summarized in section 5 (Concluding Remarks and Outlook) of the manuscript.
Reviewer 3 Report
Thank you for the response. I believe the response and changes made to the manuscript warrant the publication in this journal.
Author Response
Dear Reviewer,
Thank you for your valuable contributions that help us to improve the quality of our manuscript.